# A Fault-Signal-Based Generalizing Remaining Useful Life Prognostics Method for Wheel Hub Bearings

**Shixi Tang [1,2,\*]**, **Jinan Gu [1]**, **Keming Tang [2]**, **Rong Zou [1]**, **Xiaohong Sun [1]** and **Saad Uddin [1]**

[1] Mechanical Information Research Center, Jiangsu University, 301 Xuefu Road, Zhenjiang 212013, China; gujinan@tsinghua.org.cn (J.G.); zr@ujs.edu.cn (R.Z.); jxxsxh_1002@163.com (X.S.); saaduddin24@hotmail.com (S.U.)

[2] School of Information Engineering, Yancheng Teachers University, 50 Kaifang Avenue, Yancheng 224002, China; tkmchina@126.com

\* Correspondence: tangsx@yctu.edu.cn; Tel.: +86-515-8823-3202

**Abstract:** The goal of this work is to improve the generalization of remaining useful life (RUL) prognostics for wheel hub bearings. The traditional life prognostics methods assume that the data used in RUL prognostics is composed of one specific fatigue damage type, the data used in RUL prognostics is accurate, and the RUL prognostics are conducted in the short term. Due to which, a generalizing RUL prognostics method is designed based on fault signal data. Firstly, the fault signal model is designed with the signal in a complex and mutative environment. Then, the generalizing RUL prognostics method is designed based on the fault signal model. Lastly, the simplified solution of the generalizing RUL prognostics method is deduced. The experimental results show that the proposed method gained good accuracies for RUL prognostics for all the amplitude, energy, and kurtosis features with fatigue damage types. The proposed method can process inaccurate fault signals with different kinds of noise in the actual working environment, and it can be conducted in the long term. Therefore, the RUL prognostics method has a good generalization.

**Keywords:** data-driven method; remaining useful life prognostics; fault signal analysis; grey system; differential hydrological grey method; wheel hub bearings

---

## 1. Introduction

Wheel hub bearings are one of the key parts of automobiles. Their main function is to bear the load of the automobile and provide precise guidance for the wheel hub rotation, which means that the wheel hub bearings not only bear the axial load but also bear the radial load. To ensure maximum safety and reliability, early warning of bearing wear faults is required. Bearing wear faults include frictional noise during rotation and abnormal deceleration of the suspension wheel hub during turning. There are many moving parts and rotating parts in contact with non-rotating parts, which may generate noises. If it is confirmed that the noise comes from the wheel hub bearings, the bearing may have been damaged, and need to be replaced. The bearings play a crucial part in reliable operation of rotating machinery in manufacturing systems. There is a growing demand for smart prognostics of wheel hub bearings' remaining useful life (RUL). It is necessary to monitor the wheel hub bearings by prognosticating the RUL, especially in the initial stage of the wheel hub bearing's running stage. The best time to repair or replace the wheel hub bearings can be obtained with the RUL. Considering the aging law, degradation law, influence factors, and determining factors of wheel hub bearings, the RUL can be prognosticated to grasp the current condition and life expectancy of the wheel hub bearings. Therefore, the risk of critical accidents can be prevented and maintenance costs can be saved.

The safety hazards of the wheel hub bearings can be better removed, and the safety of the staff can be better protected too.

A number of factors influencing the fatigue life of failure of the wheel hubs of an aircraft were analyzed [1]. Experimental characterization of mechanical vibrations and acoustic noise generated by defective automotive wheel hub bearings were measured and data were analyzed with a number of signal processing methods in frequency and time-frequency domains [2]. Meng et al. proposed a fault diagnosis algorithm of the unit based on immune danger theory [3]. Ceyhan et al. discussed the wheel hub fatigue performance under non-constant rotational loading [4]. Further, the bearing fault evaluation is carried out for structural health monitoring, fault detection, failure prevention, and prognosis [5]. The measurements methods, advantages, and disadvantages of fault diagnosis methods were discussed for bearing health monitoring through shaft signals [6]. Zhao et al. proposed a deep feature optimization fusion method to extract centrifugal pump bearing degradation features from massive amounts of vibration data [7]. The multi-time scale method is used for bearing defect tracking [8]. Wang et al. described the vibration-based bearing and gear health indicators [9]. Zhou et al. investigated the possibility of a bearing health assessment method using local characteristic-scale decomposition, approximate entropy, together with manifold distance [10]. These achievements laid the foundation for bearing RUL prognostics.

The rule-based methods for bearing RUL prognostics are typically designed using Weibull model and Finite Elements. Lan et al. designed a Weibull model of the RUL for steel rebar considering corrosion effects [11]. Gastaldi et al. proposed a forced response prediction for turbine blades with flexible dampers [12]. Carlos et al. analyzed the intentional mistuning effect in the forced response of rotors with aerodynamic damping [13]. Abera et al. discussed the parameter optimization to avoid propeller-induced structural resonance of quadrotor-type unmanned aerial vehicles [14]. However, these methods have too many influence indicators, and the calculation is too complicated. The mathematical expressions are various and difficult to unify. When the equipment working environment is complex and mutative, it is hard to imply the rule and the model is also hard to design.

The data driven method for bearing RUL prognostics has shown promising potential to support smart prognostics and to avoid the design of the complicated mathematical expressions. Many data-driven methods were proposed for the RUL estimation of rolling element bearings, including support vector regression [15], sparse representation theory [16], adaptive Wiener process method [17], the time series method with Linear and nonlinear techniques [18,19], nonlinear autoregressive neural network [20,21], localized maintenance method [22], sparse representation method [23], physics-based algorithm [24], the rain flow cycle counting method [25], and grey theory [26,27]. These RUL estimation methods give static estimations of RUL.

To dynamically prognosticate the RUL, dynamical data-driven methods were proposed for further RUL prognostics of rolling element bearings. Mixed effect methods were proposed for prognostics of rolling element bearings [28]. Goodness metrics of correlation, monotonicity, and robustness are defined and combined in more relevant degradation feature selection for RUL prognostics [29]. The state–space method of linear Brownian motion was proposed to make the drift coefficient adaptive in RUL prognostics [30]. Ren et al. designed a deep-learning-based prognostics framework for bearing RUL by using a deep auto encoder and deep neural networks [31]. Jin et al. derived an auto regressive method to prognosticate its RUL according to healthy bearing vibrational signals [32]. Peng et al. proposed a data-driven method based on Gaussian mixture method and distance evaluation technique to prognosticate the RUL of rolling bearings [33]. Hu et al. proposed a performance degradation method and a real-time RUL prognostics method on the basis of temperature characteristic parameters to determine the RUL of wind turbine bearings [34]. Li et al. developed a particle filtering-based method to prognosticate the RUL of rolling element bearings [35]. Deutsch and Young designed time–frequency image features to construct health indicator (HI) and prognosticate the RUL based on continuous wavelet transformation and deep learning [36,37]. Lei et al. proposed a nonlinear degradation-based method and model-based method

for RUL prognostics of rolling element bearings [38,39]. Wang et al. designed a hybrid prognostics method for RUL prognostics of rolling element bearings [40]. Shan et al. proposed an open and data-driven-based status assessment method to obtain the individual status offset in real-time bearing residual life prognostics [41]. These RUL prognostic methods can dynamically prognosticate the RUL in real time.

The above methods focus on the RUL prognostics of bearings. For the RUL prognostics of wheel hub bearings, there are still many problems, as follows: (a) The above-mentioned methods assume that the data used in RUL prognostics is composed of one specific fatigue damage type. The actual data usually contains two or more fatigue damage types, so the specific method for specific-type fatigue damage cannot fit compound fatigue damage. (b) These methods assume that the data used in RUL prognostics is accurate. The RUL of wheel hub bearings has a great correlation with its working environment, but we cannot ensure the prognostic accuracy of RUL in complex working environments. The RUL of wheel hub bearings is affected by many factors, and the fault signals of wheel hub bearings are submerged in different kinds of noise. The premise of the above-mentioned methods is that the obtained data is accurate and has no noise. It is very difficult to obtain such data in the actual working environment. (c) These methods of RUL prognostics are conducted in the short term. However, more often, long term RUL prognostics are required to ensure maximum safety and reliability, especially in the early running stage of wheel hub bearings.

To solve these problems, it is necessary to design a generalizing RUL prognostics method. With this new method, the long term RUL of wheel hub bearings can be dynamically prognosticated in real time in a complex and changing working environment. The main novelties and contributions of this paper are as follows:

- A fault signal model is designed with the signal in a complex and mutative environment to describe the fault damage severity in the inaccurate fault signal data with different fatigue damage types.
- A generalizing RUL prognostics method is designed based on the fault signal model, and the simplified solution of the generalizing RUL prognostics method is deduced.
- The experimental results show that this method gained good accuracies of RUL prognostics for all the amplitude, energy, and kurtosis features with fatigue damage types. This method can process inaccurate fault signals with different kinds of noises in the actual working environment. Additionally, the RUL prognostics method can be conducted in the long term, so the RUL prognostics method has a good generalization.

The structure of this article is as follows. In the second part, the fault factors and indicators of wheel hub bearings are discussed. In the third part, a fault-signal-based generalizing RUL prognostics method is proposed for wheel hub bearings. The proposed RUL prognostics method is validated in the fourth part, and we draw a conclusion in the last part.

## 2. Fault Factors and Indicators of Wheel Hub Bearings

### 2.1. Fault Mechanism of Wheel Hub Bearings

The wheel hub unit is designed with an inner flange bolted to the drive shaft and an outer flange mounting the entire bearing together, as shown in Figure 1a. The wheel hub bearings are composed with an outer ring, ball, and inner ring, as shown in Figure 1b.

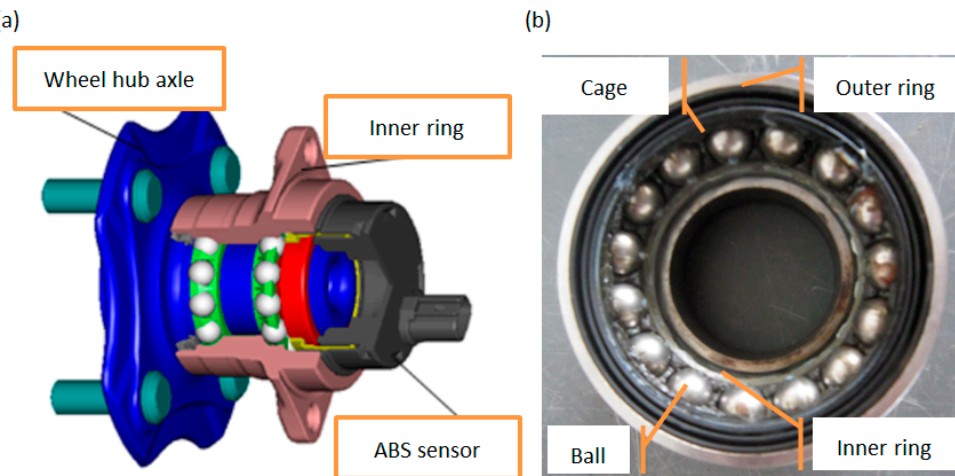

**Figure 1.** Wheel hub bearings: (**a**) wheel hub bearing unit; (**b**) wheel hub bearing structure.

In the actual operation process, the factors causing faults or even failure of the automobile hub bearing may be various, including: (a) The racking stress of the wheel hub bearings is generated by the applied load from the contact surface of the rolling element with the inner and outer ring raceways. In the maximum racking stress section, an initial defect is produced under the surface of hub bearings. Then, the racking stress evolves to the contact surface, and the surface layer begins to peel off. Eventually, a large peeling of the surface of each component of the bearing appears. (b) Due to the seamless penetration of dust and impurities, the surface of the raceway and the rolling element will cause surface wear, the clearance of the wheel hub bearing will continue to expand, and the surface roughness will increase. Then, the running accuracy of the bearing will decrease, and the corresponding operating efficiency and accuracy of the equipment will be affected. therefore, the RUL of the wheel hub bearings is reduced. (c) The plastic deformation will occur when the wheel hub bearings are directly beaten with iron tools during installation, the wheel hub bearings are subjected to a large static load or impact, an additional load is generated by thermal deformation, or the indentation is generated on the raceway surface due to hardness penetration. The plastic deformation is accompanied by severe vibration and noise. (d) The bearing has poor sealing performance during the installation process or there are corrosive materials, such as moisture, acid, or alkaline in the lubricant, which cause serious chemical corrosion of the components of the hub bearing due to chemical reactions, resulting in a decrease in running accuracy. When the bearing stops running normally, the moisture in the air condenses into water droplets due to the temperature drop, and adheres to the bearing surface, which also causes rust. (e) Serious fractures of the bearing components will occur when the applied load is greater than the maximum stress that the wheel hub bearings can suffer. At the same time, the thermal stress generated by the bearings during operation will also cause serious fractures of the bearing components. When the bearing design, manufacturing process, and assembly process are unreasonable, the ferrule rib and the roller chamfer will be missing. (f) The deformation of the cage will occur when wheel hub bearing installation is not in place, hard foreign matter infiltrates the area, or the material has defects. The friction between the inner and outer rings and the rolling may be intensified during the operation of the deformed cage, accompanied by strong vibration and noise, meaning the wheel hub bearings may be damaged or even fail earlier. All these factors will affect the results of RUL prognostics. To consider these factors together, the fault vibrational signal generated from wheel hub bearings is used in this study.

### 2.2. Fault Indicators of Wheel Hub Bearings

The vibration of wheel hub bearings mainly has three types: (1) the vibration caused by the structural characteristics of the bearing itself under normal operation; (2) the vibration generated when the bearing is abnormal due to malfunction; (3) the noise vibration of the actual working environment.

The first vibration is mainly determined by its own structure, and the second vibration reflects the fault form and damage degree of the wheel hub bearings. Their frequencies of vibration are given in the following.

### 2.2.1. Inherent Vibrational Frequency of Wheel Hub Bearings

A forced vibration is caused by the components of the wheel hub bearings when the wheel hub bearings are working in a normal condition. The inherent vibrational frequency depends on the mechanism, material, shape., and quality of the bearing itself, and is not affected by the running speed.

The inherent frequency of the rolling element is obtained in Equation (1) when the material of the bearing is steel.

$$f_b = \frac{0.424}{r}\sqrt{\frac{E}{2\rho}} \tag{1}$$

Here, $r$ is the radius of the rolling element radius (m). $E$ is the modulus of elasticity (N/m$^2$). $\rho$ is the density of material (kg/m$^3$).

The inherent frequency of the cage is obtained in Equation (2).

$$f_c = \frac{n(n^2 - 1)}{2\pi\sqrt{n^2 + 1}}a^{-2}\sqrt{\frac{EI}{M}} \tag{2}$$

where $n$ is the number of rolling elements, $a$ is the radius of the rotary axis, $I$ is the moment of inertia, and $M$ is the mass per unit length of the material.

The inherent frequency of the inner and outer rings of the wheel hub bearings is obtained in Equation (3).

$$f_n = 9.40 \times 10^5 \times \frac{h}{D^2} \times \frac{n(n^2 - 1)}{\sqrt{n^2 + 1}} \tag{3}$$

where $h$ is the thickness of the ring and $D$ is the diameter of the neutral axis of the ring.

### 2.2.2. Theoretical Fault Characteristic Frequency of Wheel Hub Bearings

When the inside of the bearing fails, such as the rolling element, inner ring, or outer ring, it causes a corresponding vibration, and its corresponding frequency is called fault characteristic frequency. This type of vibration may also be contingent. The vibration frequency of each component of the wheel hub bearing is different, resulting in different damage degrees of the bearing components. The theoretical fault characteristic frequency is obtained from the relative motion relationship between the components of the bearing.

The fault characteristic frequency of the inner raceway is obtained by Equation (4).

$$f_{bpfo} = \frac{r}{60} \times \frac{1}{2} \times n \times \left(1 - \frac{d}{D}\cos\alpha\right) \tag{4}$$

The fault characteristic frequency of outer raceway is obtained by Equation (5).

$$f_{bpfi} = \frac{r}{60} \times \frac{1}{2} \times n \times \left(1 + \frac{d}{D}\cos\alpha\right) \tag{5}$$

The fault characteristic frequency of the rolling element is obtained by Equation (6).

$$f_{bsf} = \frac{r}{60} \times \frac{1}{2} \times \frac{D}{d}\left(1 - \left(\frac{d}{D}\cos\alpha\right)^2\right) \tag{6}$$

The fault characteristic frequency of the cage is obtained by Equation (7) when the cage collides with the inner ring.

$$f_c = \frac{r}{60} \times \frac{1}{2} \times \frac{1}{n} \times \left(1 + \frac{d}{D}\cos\alpha\right) \tag{7}$$

The fault characteristic frequency of the cage is obtained by Equation (8) when the cage collides with the outer ring.

$$f_c = \frac{r}{60} \times \frac{1}{2} \times \frac{1}{n} \times \left(1 - \frac{d}{D}\cos\alpha\right) \tag{8}$$

The hub bearing speed $r$ is $r = |f_o - f_i| \times 60$, $f_o$ is the outer ring rotation frequency, $f_i$ is the inner ring rotation frequency, $n$ is the number of rolling elements, $d$ is the diameter of the rolling element. $D$ is the diameter of the bearing and $\alpha$ is the contact angle.

### 2.2.3. Feature Indicators of Wheel Bearing Vibration Signal

Vibrational signal analysis is the simplest and most effective detection method to reflect the real-time working state of the wheel hub bearing. The collected vibration signal is mapped to the intuitive and accurate time domain to identify the running situations and the damage degree of the wheel hub bearings. The signal amplitude $x_i$, signal energy $E_i$, and signal kurtosis $K_i$ are selected as the feature indicators of wheel bearing vibration signal in this research. The signal $x_i$ itself reflects the instantaneous amplitude of the fault vibrational signal over time.

The signal energy $E_i$ reflects the impact intensity of the fault vibrational signal, which is described in Equation (9). The signal energy can accurately and effectively diagnose the damage degree of the wheel hub bearing, so it is one of the most common fault diagnosis indicators in industrial production.

$$E_i = x_i^2 \tag{9}$$

The signal kurtosis $K_i$ indicator is a dimensionless parameter, described in Equation (10). T kurtosis indicator maintains high sensitivity to early bearing fault signals and is not affected by factors such as bearing size, load, and rotating shaft speed.

$$K_i = \frac{1}{N}\sum_{i=1}^{N} \frac{(|x_i| - X_m)^4}{X_{rms}^4} \tag{10}$$

Here, $X_m$ is the signal peak value computed in Equation (11). The peak value is the maximum instantaneous amplitude value of the vibration signal during a certain period of time.

$$X_m = \max\{|x_i|\} \tag{11}$$

$X_{rms}$ is the root mean square value of the signal computed in Equation (12).

$$X_{rms} = \sqrt{\frac{1}{N}\sum_{i=1}^{N} x_i^2} \tag{12}$$

## 3. Methodology

### 3.1. Fault Signal Model

The signal $x(t)$ is formed by superimposing various fault signals $e_i(t)$, environmental noise signals $n_j(t)$, and system noise signals $s_k(t)$, which is denoted by Equation (13).

$$x(t) = \sum_{i=1}^{I} e_i(t) + \sum_{j=1}^{J} n_j(t) + \sum_{k=1}^{K} s_k(t) \tag{13}$$

Here, $\sum_{i=1}^{I} e_i(t)$ is a fault signal set, $\sum_{j=1}^{J} n_j(t)$ is an environmental noise signal set, and considering that in actual operating conditions, the intensity of the various noise signals generated by the environment is very large, far stronger even than various fault signals, so $\sum_{i=1}^{I} e_i(t) << \sum_{j=1}^{J} n_j(t)$. Further, $\sum_{k=1}^{K} s_k(t)$ is a system signal set. The system signal set has relative stability. Similarly, the intensity of the various noise signals generated by the environment is stronger than the signals generated by the system itself, so $\sum_{j=1}^{J} n_j(t) > \sum_{k=1}^{K} s_k(t)$. However, a few intensities of fault signals and system signals are stronger than noise signals. The fault signals are mainly generated by the structure of wheel hub bearing interaction with the environment, reflecting both structural resonance and forced response. The noise signals are mainly generated by the environment, including the signals of the forced response. The system signals are mainly generated by the structure of the wheel hub bearing itself, including the signals of structural resonance. The fatigue characteristics are implicit in the fault signals.

To make the RUL prognostics method more effective, it is important to give prominence to the fatigue characteristics in the fault signal. We give the following principles in this study.

(a) When the external environment is the same, the severity of single fault damage is the largest for the first time, leading the severity to decrease sequentially. Therefore, Equation (13) is rewritten as Equation (14) by adding the severity factor $e^{-\alpha_i t}$.

$$x(t) = \sum_{i=1}^{I} e_i(t) \cdot e^{-\alpha_i t} + \sum_{j=1}^{J} n_j(t) + \sum_{k=1}^{K} s_k(t) \tag{14}$$

Here, $\alpha_i$ is the attenuation factor of fault signal $e_i(t)$.

(b) The cumulative fault damage severity increases gradually when the external environment is the same. Equation (14) is rewritten as Equation (15) by accumulating $e_i(\tau) \cdot e^{-\alpha_i \tau}$.

$$x(t) = \sum_{i=1}^{I} \sum_{\tau=t_0}^{t} e_i(\tau) e^{-\alpha_i \tau} + \sum_{j=1}^{J} n_j(t) + \sum_{k=1}^{K} s_k(t) \tag{15}$$

(c) The external environment always changes, so that the severity of the cumulative fault damage cannot be correctly performed in the signal $x(t)$. By the secondary accumulation of the fault signal item and the environmental noise signal item, the change of the external environment can be suppressed and the severity of the cumulative fault damage can be correctly represented. As such, Equation (14) is rewritten as Equation (16).

$$x(t) = \sum_{\tau=t_0}^{t} \left( \sum_{i=1}^{I} \sum_{\varsigma=\tau_0}^{\tau} e_i(\tau) e^{-\alpha_i \varsigma} + \sum_{j=1}^{J} n_j(\tau) \right) + \sum_{k=1}^{K} s_k(t) \tag{16}$$

The fault signal model of Equation (16) can better express the fatigue characteristics, and the RUL can be dynamically prognosticated in real time with these signal series.

*3.2. RUL Prognostics Method*

In this study, the signal containing fatigue fault information is considered as time series. Let $\overset{m}{x}$ be the discharge of the $m$-order grey process accumulative transformation of signal $x(t)$, as to the discrete sequence $\overset{m}{x}(t) = \sum_{l=1}^{t} \overset{m-1}{x}(l)$, $x(t) = \overset{0}{x}(t)$. There are $n$ values of time backtracking and $t$ samplings

in this method. The sampling periods are the same $\Delta t$. Then, the RUL prognostics method can be denoted by Equation (17) [42].

$$\overset{m}{x}(t) = \sum_{n=1}^{t} a_i \overset{m}{x}(n\Delta t) + b \tag{17}$$

Here, $a_i$ and $b$ are the system parameters of proposed method.

The RUL prognostics method with (17) can be considered as a differential hydrological grey method [22], which can be denoted by Equation (18).

$$\frac{d^n \overset{m}{x}}{dt^n} + a_1 \frac{d^{n-1} \overset{m}{x}}{dt^{n-1}} + \cdots + a_n \overset{m}{x} = b \tag{18}$$

where $\overset{m}{x}$ can be prognosticated with the solution of method in Equation (18). The RUL of wheel hub bearings can be obtained in Equation (19).

$$\overset{m-1}{x}(t) = f(\overset{m}{x}(t+1), \overset{m}{x}(t)) \tag{19}$$

The solution of the method with (17) cannot be obtained easily due to the complexity of signal $x(t)$. The simplified solution is obtained in the following.

*3.3. Method Solution*

To simplify the study work, let $m = 1$, $n = 1$, and $\Delta t = 1$. Then, Equation (17) is changed to Equation (20), and Equation (18) is changed to Equation (21).

$$\frac{d \overset{1}{x}(t)}{dt} + a \overset{1}{x}(t) = b \tag{20}$$

$$\overset{1}{x}(t) - \overset{1}{x}(t-1) + a \overset{1}{x}(t) = b \tag{21}$$

The parameters $a$, $b$ can be acquired with the least square method. The solution of RUL prognostics method in Equations (19) and (21) can be expressed as Equation (22).

$$\hat{x}(\overset{1}{t+1}) = \hat{c}e^{-\hat{a}t} + \hat{d} = [\overset{0}{x}(1) - \frac{\hat{b}}{\hat{a}}]e^{-\hat{a}t} + \frac{\hat{b}}{\hat{a}} \tag{22}$$

Considering Equation (16), $\overset{1}{x}(t)$ can be expressed as Equation (23).

$$\overset{1}{x}(t) = \sum_{l=1}^{t} \overset{0}{x}(l) = \sum_{l=1}^{t} x(l) = \sum_{l=1}^{t} (\sum_{\tau=t_0}^{l} (\sum_{i=1}^{I} \sum_{\varsigma=\tau_0}^{\tau} e_i(\tau)e^{-\alpha_i \varsigma} + \sum_{j=1}^{J} n_j(\tau)) + \sum_{k=1}^{K} s_k(k)) \tag{23}$$

Let $\rho(i, \varsigma, \alpha_i) = e^{-\alpha_i \varsigma}$. At the same time, $\rho(i, \varsigma, \alpha_i)$ is loaded into the environmental noise signal and the system noise signal. All signals are regarded as a fault signal together. Let $\overset{0}{x}(t)$ be Equation (24).

$$\overset{0}{x}(t) = \sum_{\tau=t_0}^{t} (\sum_{i=1}^{I} \sum_{\varsigma=\tau_0}^{\tau} e_i(\tau) + \sum_{j=1}^{J} n_j(\tau)) + \sum_{k=1}^{K} s_k(k) \tag{24}$$

Then, Equation (23) can be simplified to Equation (25) in Equation (24).

$$\overset{1}{x}(t) = \sum_{l=1}^{t} \rho(t, l, \alpha_l) \overset{0}{x}(l) \tag{25}$$

In Equation (25), Equation (26) can be obtained.

$$
\begin{aligned}
{}^{1}x(t) - {}^{1}x(t-1) &= \sum_{l=1}^{t} \rho(t,l,\alpha_l){}^{0}x(l) - \sum_{l=1}^{t-1} \rho(t-1,l,\alpha_{l-1}){}^{0}x(l) \\
&= \sum_{l=1}^{t-1} (\rho(t,l,\alpha_l) - \rho(t-1,l,\alpha_l)){}^{0}x(l) + \rho(t,t,\alpha_t){}^{0}x(t)
\end{aligned}
\tag{26}
$$

Equation (27) can be obtained in Equations (21) and (26).

$$
\sum_{l=1}^{t-1} (\rho(t,l,\alpha_l) - \rho(t-1,l,\alpha_l)){}^{0}x(l) + \rho(t,t,\alpha_t){}^{0}x(t) + ax{}^{1}(t) = b
\tag{27}
$$

Equation (28) can be obtained by replacing $t$ with $t+1$ in Equation (27).

$$
\sum_{l=1}^{t} (\rho(t+1,l,\alpha_l) - \rho(t,l,\alpha_l)){}^{0}x(l) + \rho(t+1,t+1,\alpha_t){}^{0}x(t+1) + ax{}^{1}(t+1) = b
\tag{28}
$$

Equation (29) can be obtained by replacing $t$ with $t+1$ in Equation (21).

$$
x{}^{1}(t+1) - x{}^{1}(t) + ax{}^{1}(t+1) = b
\tag{29}
$$

Equation (29) can be obtained in Equation (30).

$$
b - ax{}^{1}(t+1) = \hat{x}{}^{1}(t+1) - \hat{x}{}^{1}(t)
\tag{30}
$$

Equation (31) is obtained by substituting Equation (30) into Equation (28).

$$
\hat{x}{}^{0}(t+1) = \rho^{-1}(t+1,t+1,\alpha_t)\cdot[\hat{x}{}^{1}(t+1) - \hat{x}{}^{1}(t) + \sum_{l=1}^{t} (\rho(t,l,\alpha_l) - \rho(t+1,l,\alpha_l)){}^{0}x(l)]
\tag{31}
$$

Equations (22) and (31) are the solution of the simplified RUL prognostics method in Equation (20). The RUL of wheel hub bearings can be obtained in Equation (31). In Equation (31), the $(t+1)^{th}$ signal $x(t+1)$ can be prognosticated with the signals from $x(1)$ to $x(t)$; $\rho(t,l,\alpha_l)$ reflects the impact of past historical signals on prognostics, called memory item. The attenuation factor $\alpha_l$ reflects the attenuation of the influence of the past signal on the prognostics. In this study, Equation (31) will be used to verify the experiment in Section 4.

This proposed RUL prognostics method only utilizes the information of fault signal, so directly obtaining parameters of the fracture mechanics can be avoided. The RUL can be dynamically prognosticated in real time with respect to time change. When the amplitude of the prognosticated signal $x(t+1)$ reaches the given maximum value, the wheel hub bearings can be considered damaged, and time $t$ is the RUL of the wheel hub bearings. The proposed RUL prognostics method can prognosticate the RUL with the information from the fault signal in the initial running stage, which is verified in Section 4.

## 4. Validation

### 4.1. Experimental Devices and Sensors

The experiment is implemented in the LGM-15/45 wheel hub bearing fatigue testing rigs using mud, as shown in Figure 2. The main body of the testing rig adopts a cantilever structure. The test bearing is placed at the cantilever end of the main shaft, and the connection between the test main shaft and the drive shaft adopts the method of stop positioning and flange connection. This structure has strong simulation ability. This structure is also convenient in assembly and disassembly.

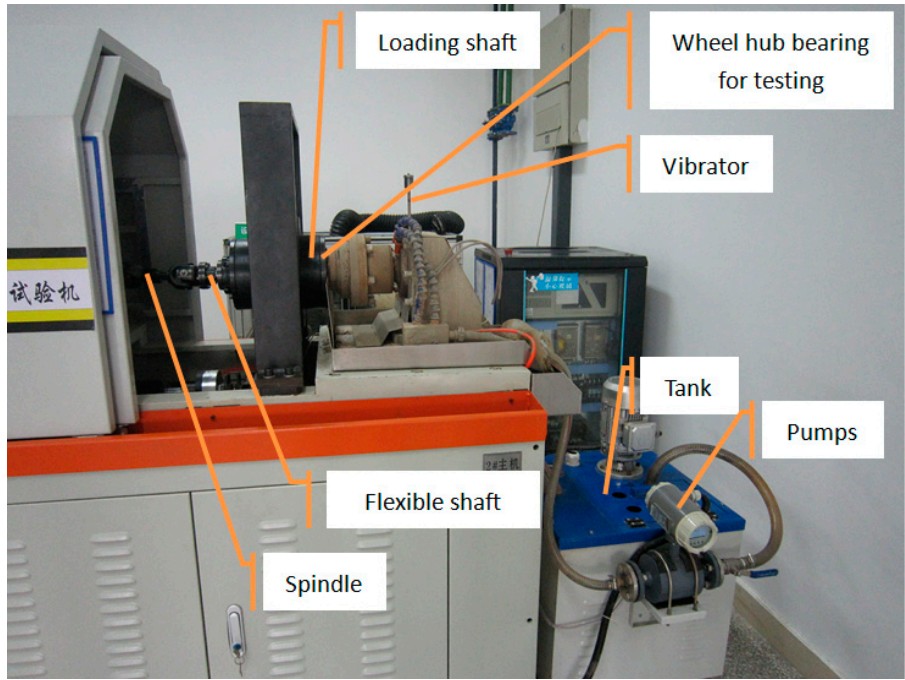

**Figure 2.** The wheel hub bearing fatigue test rigs using mud.

The transmission system in the rigs controls the AC motor by the frequency converter. The belt drive is used to drive the bearing rotation, which facilitates the step-less speed regulation of the transmission system. The loading system uses the electro-hydraulic proportional system. The electro-hydraulic proportional loading system is a closed loop control system. The system has good dynamic performance, has fast response, and is stable within ±0.01 kN.

The speed control system adopts the microcomputer control VFD-A05543B intelligent frequency converter, which can switch the inverter Insulate-Gate Bipolar Transistor (IGBT) at high speed. It has multiple automatic protection functions, which can self-test current, voltage, and frequency. The rigs communicate with the microcomputer through the RS-485 serial port by the microcomputer output signal.

The loading control system selects the BLR-42 load sensor with tension and compression style. The signal of 0–5 V is given by the microcomputer, and the given signal enters the proportional controller to continuously output the specified oil pressure under the action of the proportional amplifier and applies load to the test bearing by pushing the cylinder. This signal is converted into a voltage signal by the load sensor and the transmitter. The voltage signal is sent to the microcomputer to form a closed loop control. The system has a large control range and high precision, and can control the required load of the test bearing at a long distance and is continuously proportional.

The temperature control system controls the temperature by using the temperature sensor with a semiconductor silicon PN junction. The temperature sensor changes the current of the heater through controlling the conduction angle of the triac with a voltage regulator.

The vibration signal is collected with a model TD-3 accelerometer installed at the test point. The signal is amplified by the HD-7 charge amplifier to get the voltage value. The voltage value is input into the computer and processed according to the digital mode to reflect the amplitude of bearing vibration.

The rigs adopt microcomputer automatic control and automatic test technologies. The test bearing types include tapered roller bearings, angular contact ball bearings, deep groove ball bearings, and wheel hub bearing units. The rotation modes include inner ring and outer ring rotation. The size of the test bearing inner diameter is 15–45 mm. The axial maximum load is ±15 kN. The test bearing

maximum speed is 3000 r/min, and the measurement parameters include speed, load, temperature, vibration, and current.

In the experiment, the rotating speed is 900 rpm. The media ratios are 10% loess, 84% water, and 6% sand with F100#. The water spray speed is 5 L/min. The axial pressure is 2.667 kN and the radial load is 4.445 kN. The test object is the front wheel hub bearings of a DAC4007440 automobile. The sampling rate is 6000.

### 4.2. Data Obtainment

The waveform of the wheel hub bearing signal is shown in Figure 3. At the last moment of the vibration signal stream, the experimental instrument activated an alarm; the actual fault was occurred after 17 h and 55 min. After the fault occurred, we checked the damaged hub bearing and found that the fault occurred in the inner hub bearing. The rolling contact angle $\alpha$ is 70.12$^\circ$. The hub bearing speed $r$ is 300 rev/min. The bearing outer diameter $D_r$ is 52 mm. The bearing diameter $D_i$ is 25 mm. The number of balls $n$ is 15. The diameter of the rolling element $d$ is 25.4 mm. The bearing pitch diameter $D$ can be calculated from Equation (32). The actual frequencies of the theoretical values are $f_{bpfo} = 21.3$ Hz and $f_{bpfi} = 45.91$ Hz, calculated with Equations (1)–(8).

$$D = \frac{D_r + D_i}{2} = 38.5 \text{ mm} \tag{32}$$

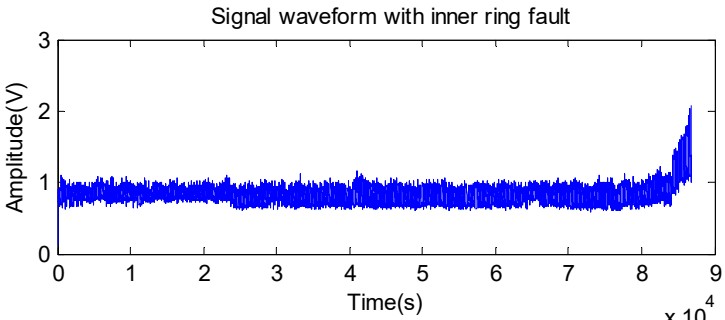

**Figure 3.** The waveform of wheel hub bearing signal.

The Fast Fourier Transformation (FFT) spectrum of the wheel hub bearing signal is shown in Figure 4. The theoretical fault frequency values $f_{bpfo} = 21.3$ Hz and $f_{bpfi} = 45.91$ Hz can be detected in the FFT spectrum. Due to the influence of various fault noise signals described in Equation (13), the fault frequency $f_{bpfo} = 21.3$ Hz is submerged in the noise signals. However, the fault frequency $f_{bpfi} = 45.91$ Hz is still obvious in the FFT spectrum.

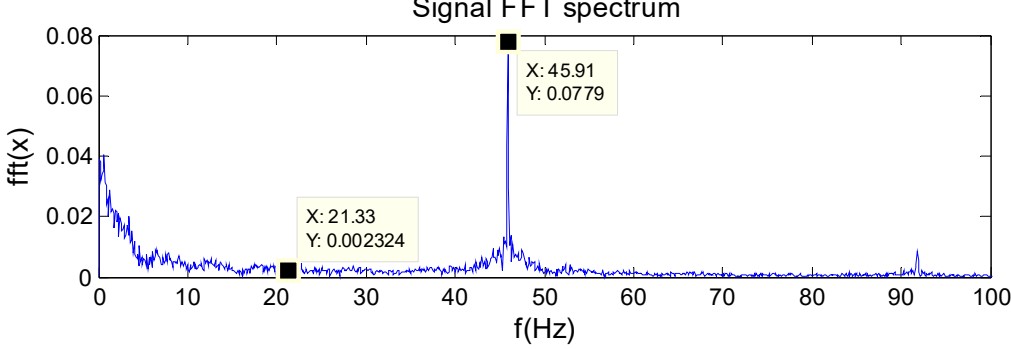

**Figure 4.** The FFT spectrum of the wheel hub bearing signal.

### 4.3. Analysis of Experimental Results

As a comparison, the RUL of wheel hub bearings is prognosticated utilizing traditional grey theory [27] with original signal (GM). The original signal is described in Equation (13). For the amplitude feature, the threshold is the maximum amplitude value of the original signal when the damage happens. Here, the amplitude threshold value is taken as 1.4. The RUL prognostics results are shown in Figure 5 and Table 1. The accuracies of RUL prognostics are 0.6429, 0.5571, 0.5642, and 0.6214 based on 25%, 50%, 75%, and 100% of the original data, as shown in Figure 5a–d. It is found that the prognostics results are random. The accuracy reduces as the original data increases for some prognostics results, which indicate that the traditional grey theory cannot prognosticate the RUL of wheel hub bearings with the original signal in a complex working environment.

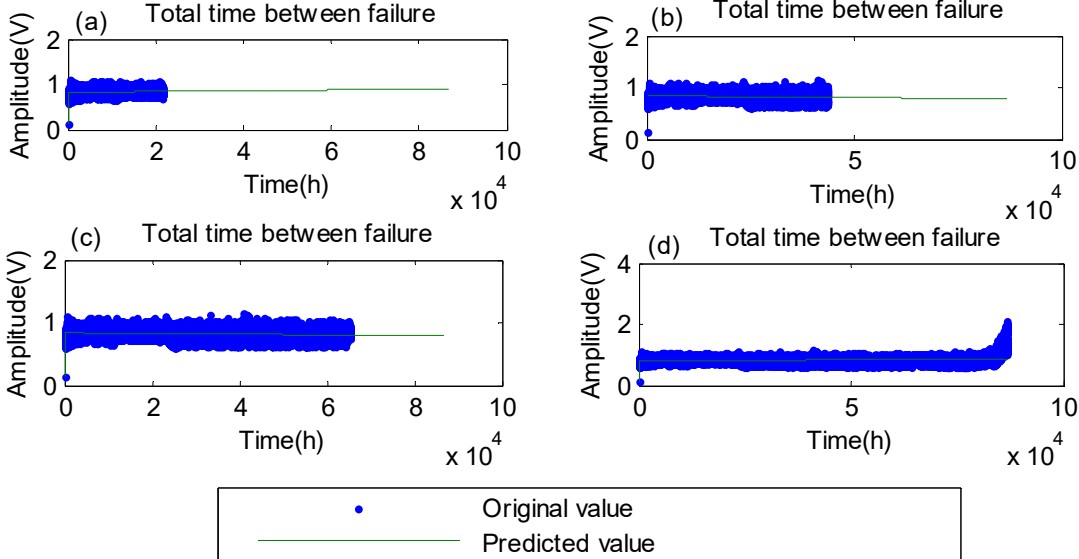

**Figure 5.** The remaining useful life (RUL) prognostics results with amplitude of original signal utilizing grey theory: (**a**) prognostics result based on 25% of the original data; (**b**) prognostics result based on 50% of the original data; (**c**) prognostics result based on 75% of the original data; (**d**) prognostics result based on 100% of the original data.

For the energy feature with Equation (9), the threshold value is taken as 0.12. The RUL prognostics results are shown in Figure 6 and Table 2. The accuracies of RUL prognostics are 0.6648, 0.5215, 0.5279, and 0.6759 based on 25%, 50%, 75%, and 100% of the original data, as shown in Figure 6a–d. It is found that the prognostics results are also random. Therefore, the prognostics results indicate that the traditional grey theory cannot prognosticate the RUL of wheel hub bearings with the original signal in a complex working environment either.

For the kurtosis feature with Equation (10), the threshold value is taken as 1.2. The RUL prognostics results are shown in Figure 7 and Table 3. The accuracies of RUL prognostics are 0.5026, 0.3511, 0.6586, and 0.7264 based on 25%, 50%, 75%, and 100% of the original data, as shown in Figure 7a–d. It is found that the prognostics results are also random. The prognostics results indicate that the traditional grey theory also cannot prognosticate the RUL of wheel hub bearings with the original signal in a complex working environment.

The RUL of wheel hub bearings can be prognosticated utilizing traditional grey theory with accumulative signal (GA). The threshold is the maximum time when the damage happens. Here, the time threshold value is taken as $8.4 \times 10^4$ h. For the amplitude feature, the RUL prognostics results are shown in Figure 8 and Table 1. The accuracies of RUL prognostics are 0.4405, 0.6905, 0.8810, and 0.9988 based on 25%, 50%, 75%, and 100% of the accumulative data, as shown in Figure 8a–d. The accuracy increases as the original data increases, which indicates that the traditional grey theory

can prognosticate the RUL of wheel hub bearings with the accumulative signal in a complex and variable working environment. However, the accuracy of RUL prognostics is very poor in the early stage with original data less than 50%. Therefore, the traditional grey theory has a poor generalization with accumulative signal.

For the energy feature, the RUL prognostics results are shown in Figure 9 and Table 2. The accuracies of RUL prognostics are 0.4357, 0.7040, 0.8910, and 0.9998 based on 25%, 50%, 75%, and 100% of the accumulative data, as shown in Figure 9a–d. Similarly, the accuracy increases as the original data increases. To some extent, the traditional grey theory can prognosticate the RUL of wheel hub bearings with the accumulative signal in a complex and variable working environment. The accuracy of RUL prognostics is also very poor in the early stage. This verifies that the traditional grey theory also has a poor generalization with the accumulative signal.

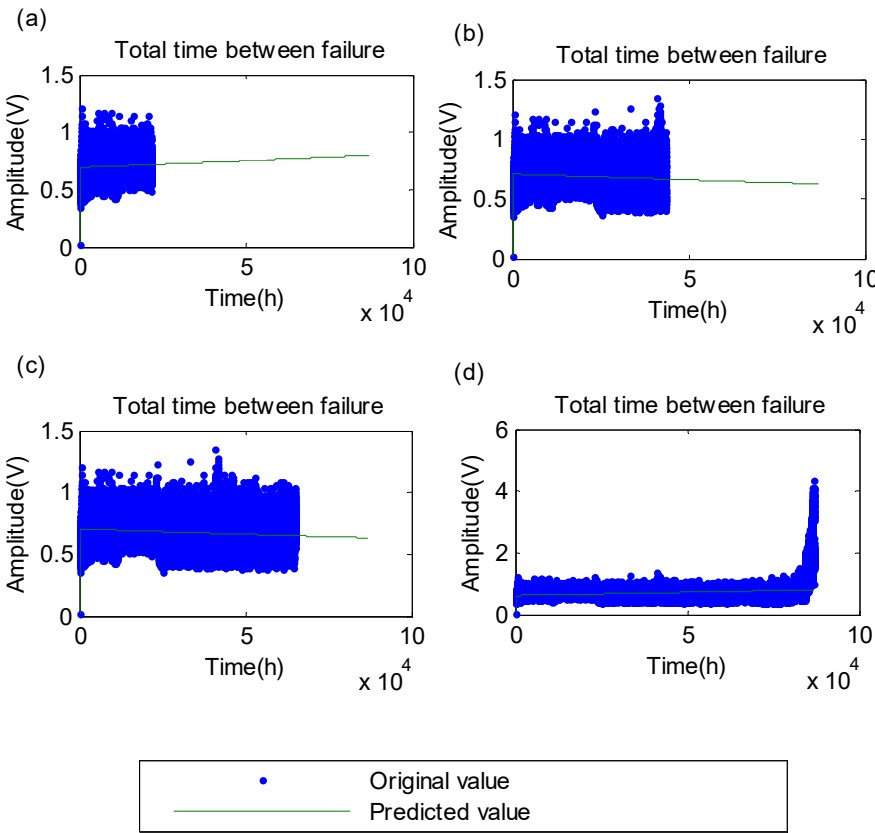

**Figure 6.** The RUL prognostics results with the energy of the original signal utilizing grey theory: (**a**) prognostics result based on 25% of the original data; (**b**) prognostics result based on 50% of the original data; (**c**) prognostics result based on 75% of the original data; (**d**) prognostics result based on 100% of the original data.

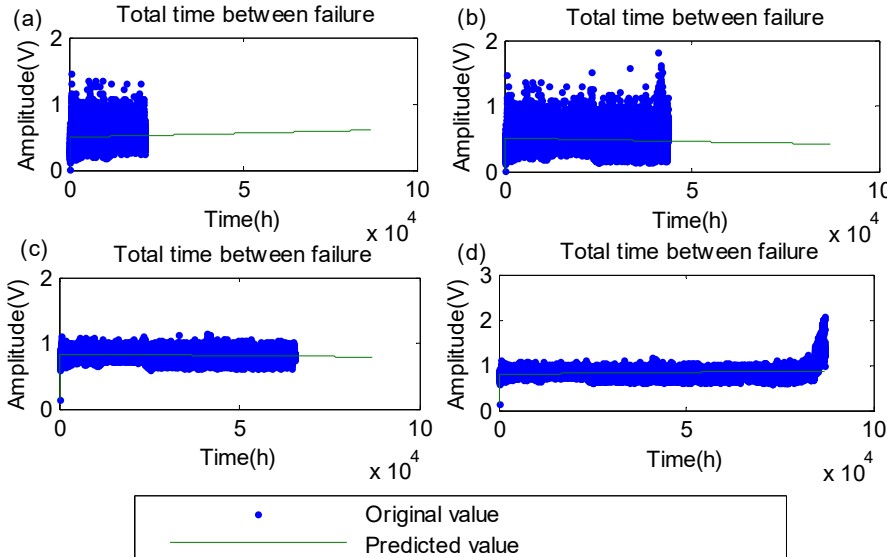

**Figure 7.** The RUL prognostics results with kurtosis of the original signal utilizing grey theory: (**a**) prognostics result based on 25% of the original data; (**b**) prognostics result based on 50% of the original data; (**c**) prognostics result based on 75% of the original data; (**d**) prognostics result based on 100% of the original data.

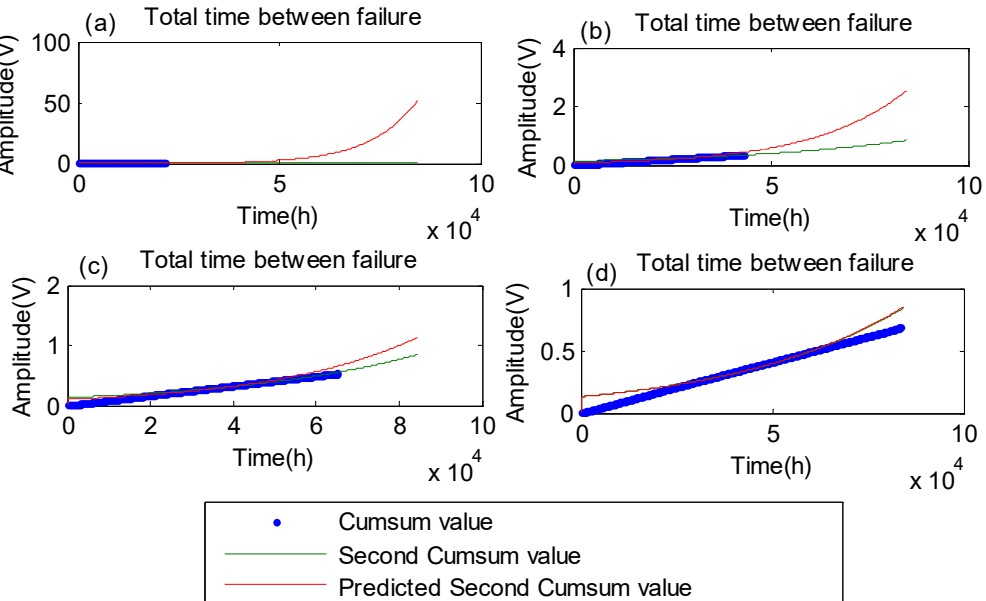

**Figure 8.** The RUL prognostics results with the amplitude of the accumulative signal utilizing grey theory: (**a**) prognostics result based on 25% of the accumulative data; (**b**) prognostics result based on 50% of the accumulative data; (**c**) prognostics result based on 75% of the accumulative data; (**d**) prognostics result based on 100% of the accumulative data.

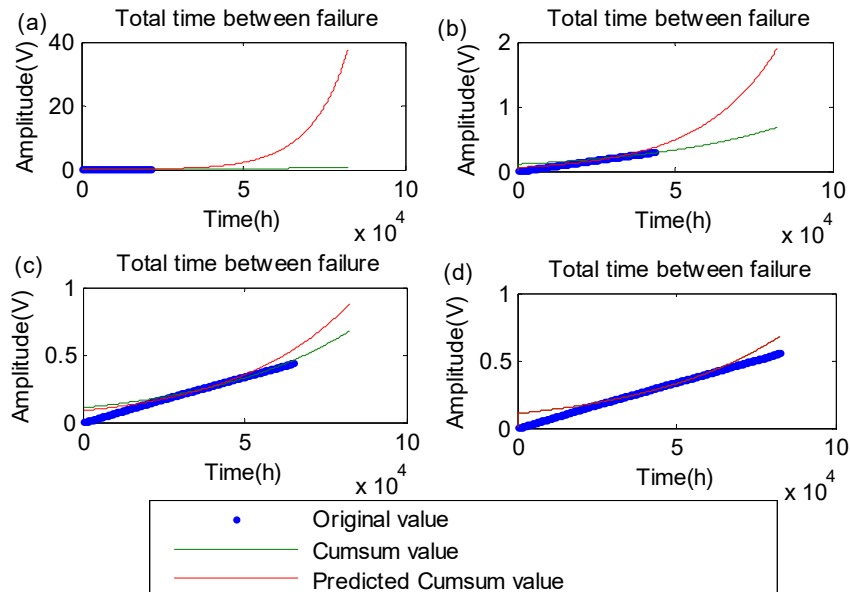

**Figure 9.** The RUL prognostic results with the energy of the accumulative signal utilizing grey theory: (**a**) prognostics result based on 25% of the accumulative data; (**b**) prognostics result based on 50% of the accumulative data; (**c**) prognostics result based on 75% of the accumulative data; (**d**) prognostics result based on 100% of the accumulative data.

For the kurtosis feature, the RUL prognostics results are shown in Figure 10 and Table 3. The accuracies of RUL prognostics are 0.4280, 0.6924, 0.8762, and 0.9909 based on 25%, 50%, 75%, and 100% of the accumulative data, as shown in Figure 10a–d. The accuracy also increases as the original data increases. The traditional grey theory can prognosticate the RUL of wheel hub bearings in a way. The accuracy of RUL prognostics is also very poor in the early stage. This also verifies that the traditional grey theory has a poor generalization with the accumulative signal.

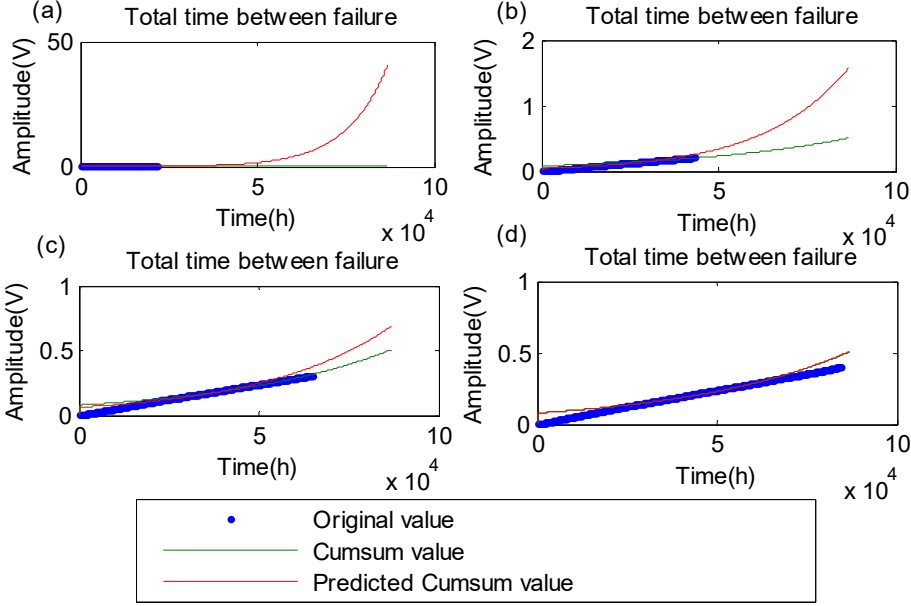

**Figure 10.** The RUL prognostics results with kurtosis of the accumulative signal utilizing grey theory: (**a**) prognostics result based on 25% of the accumulative data; (**b**) prognostics result based on 50% of the accumulative data; (**c**) prognostics result based on 75% of the accumulative data; (**d**) prognostics result based on 100% of the accumulative data.

The RUL of wheel hub bearings is prognosticated utilizing the proposed RUL prognostics method based on the fault signal with the accumulative signal (GG). The original signal is described in Equation (16). The threshold is the maximum time when the damage happens. Here, the time threshold value is taken as $8.67 \times 10^4$ h. The $\alpha_t$ in Equation (31) is calculated with Equation (33).

$$\alpha_t = \frac{t}{T_{RUL}} \times 100\% \tag{33}$$

$T_{RUL}$ is a constant. Its value only influences the slope of the prognostics curves, but has no influence on the accuracy of RUL. Here, the value is taken as $8.4 \times 10^4$.

For the energy feature, the RUL prognostics results are shown in Figure 11 and Table 1. The accuracies of RUL prognostics are 0.6113, 0.8708, 0.9804, and 0.9997 based on 25%, 50%, 75%, and 100% of the accumulative data, as shown in Figure 11a–d. The accuracy increases as the original data increases, which indicates that the proposed method can prognosticate the RUL of wheel hub bearings with the accumulative signal in a complex and variable working environment. The RUL prognostics have relatively high precision in the initial stage with original data less than 50%. Therefore, the proposed method has a good generalization with the accumulative signal.

For the energy feature, the RUL prognostics results are shown in Figure 12 and Table 2. The accuracies of RUL prognostics are 0.6160, 0.8788, 0.9880, and 0.9993 based on 25%, 50%, 75%, and 100% of the accumulative data, as shown in Figure 12a–d. The accuracy increases as the original data increases, and the RUL prognostics have relatively high precision in the initial stage. This also verifies that the proposed method has a good generalization with the accumulative signal.

For the kurtosis feature, the RUL prognostics results are shown in Figure 13 and Table 3. The accuracies of RUL prognostics are 0.6050, 0.8817, 0.9976, and 0.9996 based on 25%, 50%, 75%, and 100% of the accumulative data, as shown in Figure 13a–d. The results show that the proposed method has a good generalization with the accumulative signal.

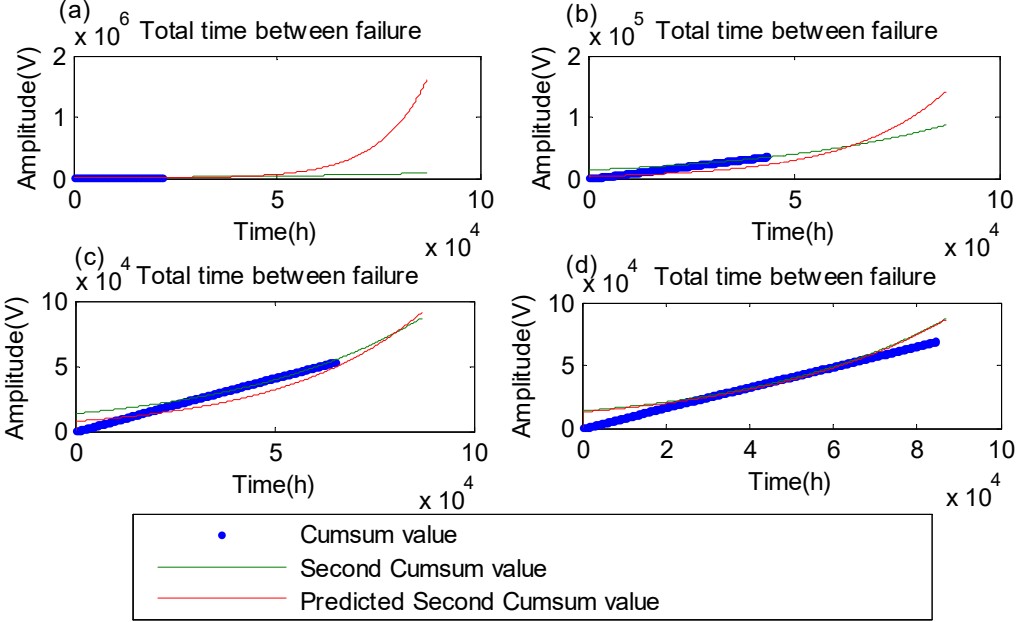

**Figure 11.** The RUL prognostics results with amplitude of the accumulative signal utilizing the proposed method: (**a**) prognostics result based on 25% of the accumulative data; (**b**) prognostics result based on 50% of the accumulative data; (**c**) prognostics result based on 75% of the accumulative data; (**d**) prognostics result based on 100% of the accumulative data.

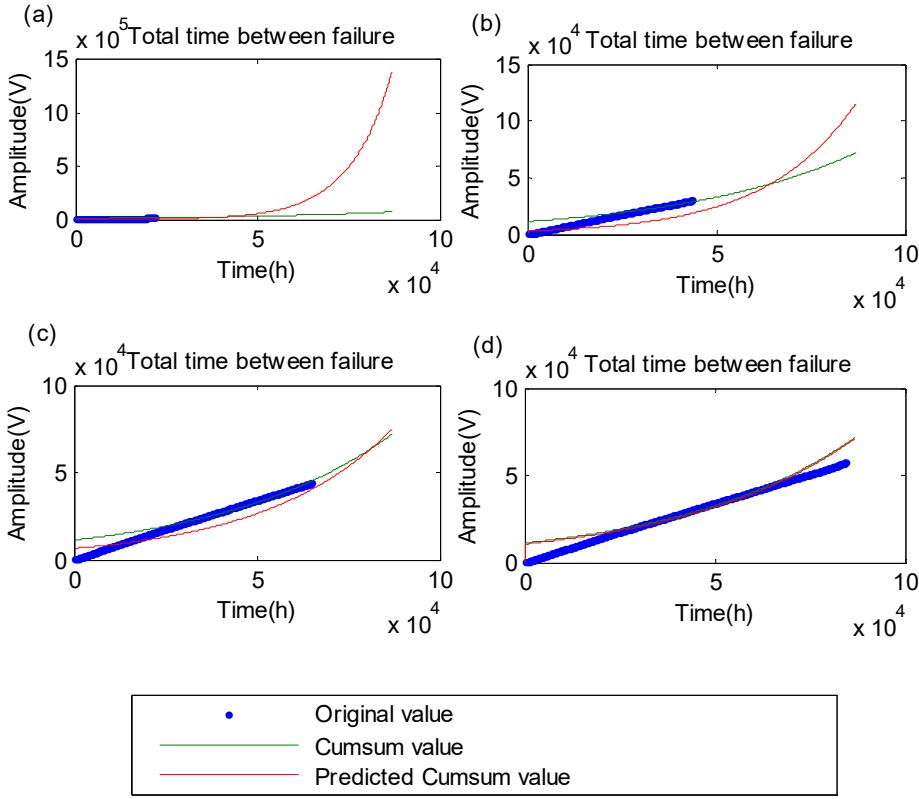

**Figure 12.** The RUL prognostics results with the energy of the accumulative signal utilizing the proposed method: (**a**) prognostics result based on 25% of the accumulative data; (**b**) prognostics result based on 50% of the accumulative data; (**c**) prognostics result based on 75% of the accumulative data; (**d**) prognostics result based on 100% of the accumulative data.

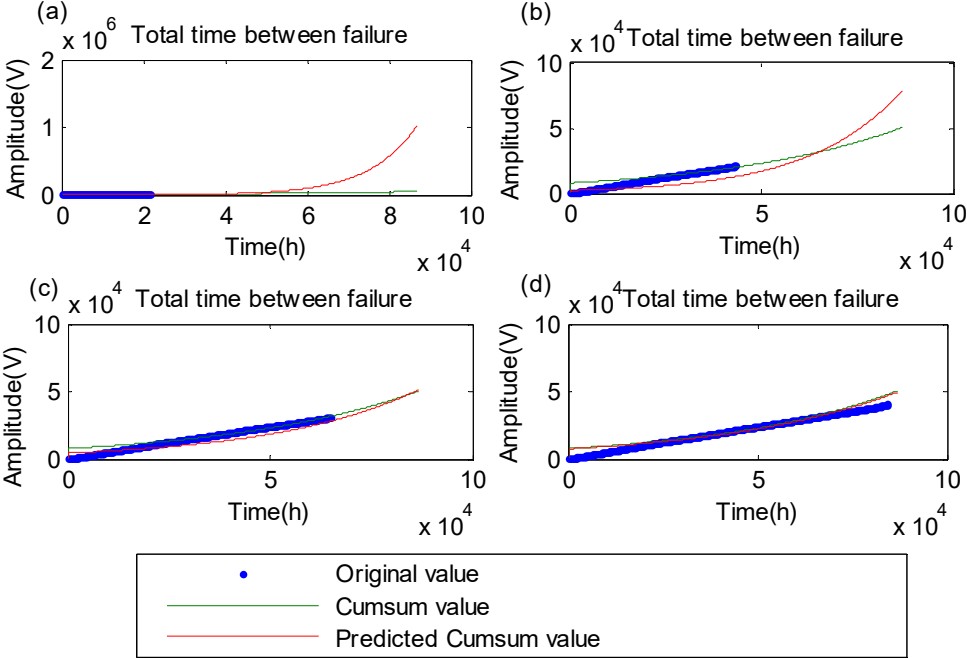

**Figure 13.** The RUL prognostics results with kurtosis of the accumulative signal utilizing the proposed method: (**a**) prognostics result based on 25% of the accumulative data; (**b**) prognostics result based on 50% of the accumulative data; (**c**) prognostics result based on 75% of the accumulative data; (**d**) prognostics result based on 100% of the accumulative data.

**Table 1.** The accuracy of RUL prognostics with the amplitude feature.

| Accuracy | *Original Data* (25%) | *Original Data* (50%) | *Original Data* (75%) | *Original Data* (100%) |
|---|---|---|---|---|
| GM | 0.6429 | 0.5571 | 0.5642 | 0.6214 |
| GA | 0.4405 | 0.6905 | 0.8810 | 0.9988 |
| GG | 0.6113 | 0.8708 | 0.9804 | 0.9997 |

**Table 2.** The accuracy of RUL prognostics with the energy feature.

| Accuracy | *Original Data* (25%) | *Original Data* (50%) | *Original Data* (75%) | *Original Data* (100%) |
|---|---|---|---|---|
| GM | 0.6648 | 0.5215 | 0.5279 | 0.6759 |
| GA | 0.4357 | 0.7040 | 0.8910 | 0.9998 |
| GG | 0.6160 | 0.8788 | 0.9880 | 0.9993 |

**Table 3.** The accuracy of RUL prognostics with the kurtosis feature.

| Accuracy | *Original Data* (25%) | *Original Data* (50%) | *Original Data* (75%) | *Original Data* (100%) |
|---|---|---|---|---|
| GM | 0.5026 | 0.3511 | 0.6586 | 0.7264 |
| GA | 0.4280 | 0.6924 | 0.8762 | 0.9909 |
| GG | 0.6050 | 0.8817 | 0.9976 | 0.9996 |

From the above analysis, we can see that: (a) Compared with GM and GA, the proposed method GG gains similar satisfactory accuracies of RUL prognostics for all the amplitude, energy, and kurtosis features with fatigue damage types. (b) The proposed method GG can process inaccurate fault signals with different kinds of noise in the actual working environment. (c) The proposed method GG has good accuracy of RUL prognostics in the early and middle stages of the wheel hub bearing's running stage, that is, the proposed method GG of RUL prognostics is conducted in the long term, which ensures maximum safety and reliability, especially in the early stage of the wheel hub bearing's running stage. As can be seen from the above, the proposed method GG has good generalization.

## 5. Conclusions

The RUL prognostics method designed in this study only utilizes the information of the fault signal. This method successfully avoids the problem of directly obtaining parameters of fracture mechanics in complex and mutative environments during the working process. The RUL can be dynamically prognosticated in real time with respect to time change for inaccurate fault signal with fatigue damage types in the actual working environment. This method also has the ability to obtain better accuracy for prognostics results, especially in the wheel hub bearing's initial running stage. The proposed method gained good accuracies for RUL prognostics for the amplitude, energy, and kurtosis features with fatigue damage types. The proposed method can process inaccurate fault signals with different kinds of noises in an actual working environment. The proposed RUL prognostics method has better long term prognostics ability than existing methods. As such, the proposed RUL prognostics method has good generalization.

The current method is only the simplified solution of the generalizing RUL prognostics method in Equation (18), combined with the fault signal model in Equation (16). To further improve the accuracy of RUL prognostics results, (a) a relatively complete solution, as in Equations (16) and (18), should be studied in future work, and (b) the fault signals should be decomposed from the synthesis signal with fault signals, environmental noise signals, and system noise signals in future work.

**Author Contributions:** S.T. and J.G. conceived the idea and formulated the mathematical methods behind the provided method. K.T. was responsible for all parameter tuning, data processing, figures, and calculations. R.Z. was responsible for the experimental setup. X.S. and S.U. were responsible for research idea generation, the introduction, abstract, conclusions, and for the final editing and reviewing of the provided method.

**Funding:** The work was funded by the National Science Foundation of China [51875266, 61379064, 61603326, 61772448]; the Industry-Academia Prospect Research Foundation of Jiangsu [BY2016066-06, BY2015058-01]; the Research Innovation Program for College Graduates of Jiangsu [KYLX16_0880].

**Conflicts of Interest:** The authors declare no conflict of interest.

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
