# Peer review of "A Fault-Signal-Based Generalizing Remaining Useful Life Prognostics Method for Wheel Hub Bearings"

_applsci, doi:10.3390/app9061080_

Reviewer 1 Report

It sounds an interesting work and I have below questions for the authors to consider:

1 Do you think the developed method could reveal the "structural resonance" in the early stage?

2 After the early stage, due to the development of the fault, do you think the "structural resonance" will be replaced by the "forced response"?

3 Do you think in your RUL model, above questions should be reflected?

Author Response

Dear Reviewer: 
We wish to thank you for the time and effort you have spent reviewing our paper. We are pleased to note that you have found our research work interesting and also pointed out some problems to help us improve the quality of our work. 
Motivated by your comments, we have deeply reconsidered the architecture of our work and tried to fix all the problems you mentioned. In particular, this revised manuscript of our resubmitted letter has significantly been improved mainly as follows: (The improvements are in red in the revised version using the "Track Changes" function in Microsoft Word for your convenience.)

Thank you for your interesting and important questions. Our responses to the questions are as follows.

1. Do you think the developed method could reveal the "structural resonance" in the early stage?

R: Our developed method focuses on the remaining useful Life prognostics based on fault signal. The structural resonance signals of wheel hub bearings in the early stage are only implicit in the fault signals and system signals. We do not reveal the structural resonance independently.

2. After the early stage, due to the development of the fault, do you think the "structural resonance" will be replaced by the "forced response"?

R: We think that the forced response will become much stronger due to the development of the fault. But the structural resonance still exists.

3. Do you think in your RUL model, above questions should be reflected?

R: Our developed RUL model is a data driven model, which is different from the rule based model, as explained in lines 59-76. The data driven model emphasizes utilizing the signal data itself, and above questions are only reflected in an implicit way, as explained in lines 244-255 in the revised version.

The questions you proposed are very meaningful. We are designing a rule based RUL model in our current work, and we will fully consider your opinion in this work. Thanks again.

Reviewer 2 Report

Overall the paper presents high quality of research. The research topic is interesting to readers. Especially the useful life prediction of rotational components based on dynamic fault signal processing attracts attentions to many researchers in different fields.  

In section 4. Validation, the experimental results are displayed clearly. However, the presentation of the research results is a bit lengthy because of a lot of repeated information. If the repeated information can be presented in a little bit concise way, that would be better. 

Author Response

Dear Reviewer: 
We wish to thank you for the time and effort you have spent reviewing our paper. We are pleased to note that you have found our research work interesting and also pointed out some problems to help us improve the quality of our work. 
Motivated by your comments, we have deeply reconsidered the architecture of our work and tried to fix all the problems you mentioned. In particular, this revised manuscript of our resubmitted letter has significantly been improved mainly as follows: (The improvements are in red in the revised version using the "Track Changes" function in Microsoft Word for your convenience.)

1. In section 4. Validation, the experimental results are displayed clearly. However, the presentation of the research results is a bit lengthy because of a lot of repeated information. If the repeated information can be presented in a little bit concise way, that would be better.

R: The repeated information of the research results is presented in a concise way, as shown in lines 401-402, lines 407-409, lines 415-417, lines 423-425, lines 430-432,lines 445-447, lines 460-462, lines 467-469 and lines 472-474 in the revised version.

Reviewer 3 Report

This paper deals with an interesting and important topic, the improvement of the generalization of remaining useful life (RUL) prognostics, in this case applied to wheel hub bearings. Although the idea is good, since this is a very interesting research area, the work has several points to improve, as detailed below.

1.- English grammar and style should be revised all along the manuscript.

2.- The authors must provide a detailed explanation about the novelties and contributions of this paper beyond the state of the art. To this end it would be nice a paragraph like this one: “The main novelties and contributions of this paper are as follows: …”

3.- Figure 1. Please increase the size of the fonts as well as the resolution of Figure 1b.

4.- The conditions or comparisons made in the paragraph below equation (13) should be further detailed (lines 219-221).

5.- Line 219: “The media ratios are 10% of loses...” Which kind of loses do you refer? Water loses in the system?

6.- Figure 3. The authors plot voltage against time to detect the fault. I guess that the voltage signal comes from a sensor. The characteristics of this sensor must be given, and a clear justification of the motivation of using such sensor is required since all results in this paper are based on this amplitude signal.

7.- I assume it is possible to perform an FFT of the signal acquisitions shown in Figure 3. Could the authors check if the theoretical frequencies calculated in line 312 are corroborated in the FFT spectrum? It is important since it is a way to corroborate the usefulness of your measurement system.

8.- It would be nice to include a subsection describing in detail all instruments, sensors and devices used in the experimental, providing the main characteristics of all them.

The Reviewer aims the authors to revise the work based on the suggestions above in order to improve its quality.

Author Response

Dear Reviewer: 
We wish to thank you for the time and effort you have spent reviewing our paper. We are pleased to note that you have found our research work interesting and also pointed out some problems to help us improve the quality of our work. 
Motivated by your comments, we have deeply reconsidered the architecture of our work and tried to fix all the problems you mentioned. In particular, this revised manuscript of our resubmitted letter has significantly been improved mainly as follows: (The improvements are in red in the revised version using the "Track Changes" function in Microsoft Word for your convenience.)

1. English grammar and style should be revised all along the manuscript.

R: We revised the English grammar and style all along the manuscript.

2. The authors must provide a detailed explanation about the novelties and contributions of this paper beyond the state of the art. To this end it would be nice a paragraph like this one: “The main novelties and contributions of this paper are as follows: …”

R: A detailed explanation about the novelties and contributions of this paper is provided in lines 115-124 in the revised version.

3. Figure 1. Please increase the size of the fonts as well as the resolution of Figure 1b.

R: The size of the fonts as well as the resolution of Figure 1b is increased in line 166 in the revised version.

4. The conditions or comparisons made in the paragraph below equation (13) should be further detailed (lines 219-221).

R: The conditions or comparisons made in the paragraph below equation (13) are further detailed in lines 244-255 in the revised version.

5. Line 219: “The media ratios are 10% of loses...” Which kind of loses do you refer? Water loses in the system?

R: We are sorry that the “loess” is misspelled as “loses”. We replaced “The media ratios are 10% of loses...” with “The media ratios are 10% of loess...” in line 372 in the revised version.

6. Figure 3. The authors plot voltage against time to detect the fault. I guess that the voltage signal comes from a sensor. The characteristics of this sensor must be given, and a clear justification of the motivation of using such sensor is required since all results in this paper are based on this amplitude signal.

R: The characteristics of sensors and the motivation of using these sensors are given in lines 389-393 in the revised version.

7. I assume it is possible to perform an FFT of the signal acquisitions shown in Figure 3. Could the authors check if the theoretical frequencies calculated in line 312 are corroborated in the FFT spectrum? It is important since it is a way to corroborate the usefulness of your measurement system.

R: The theoretical frequencies calculated in line 312 are corroborated in the FFT spectrum as shown in Figure 4 and are explained in lines 389-395 in the revised version.

8. It would be nice to include a subsection describing in detail all instruments, sensors and devices used in the experimental, providing the main characteristics of all them.

R: We added a subsection describing in detail all instruments, sensors and devices used in the experimental, providing the main characteristics of all them in lines 334-371 in the revised version.

Round  2

Reviewer 1 Report

Thanks for your consideration.

Reviewer 3 Report

The authors have replied all my questions